# Experimental study on pore structure characteristics and thermal conductivity of fibers reinforced foamed concrete

Lin Li[1,2], Wei Wang[1], Yu Wang[1,2], Dongxu Li[3], Mei-Ling Zhuang[4]*

1 School of Architectural Engineering, Suqian College, Suqian, China, 2 Jiangsu Province Engineering Research Center of Prefabricated Building and Intelligent Construction, Suqian, China, 3 College of Material Science and Engineering, Nanjing Tech University, Nanjing, China, 4 School of Transportation and Civil Engineering, Nantong University, Nantong, China

* ml_zhuang99@163.com

## Abstract

The pore structure characteristics and thermal conductivity of foamed concrete (FC) reinforced with glass fibers (GF), polyvinyl alcohol fibers (PVAF) and polypropylene fibers (PPF) were investigated experimentally in this article. Firstly, GF, PVAF or PPF with different mass fractions (0%, 1%, 1.5% and 2%) were added to the Portland cement, fly ash and plant protein foaming agent to prepare the FC. Then, SEM tests, dry density tests, porosity tests, and thermal conductivity tests were carried out on FRFC. Later, the adhesion of GF, PVAF and FFF with different mass fractions to the cementitious base was investigated by SEM images of FRFC. The pore size distribution, shape factor and porosity of FRFC were analyzed using Photoshop software and Image Pro Plus (IPP) software. Finally, the effects of different mass fractions and lengths of three types of fibers on the thermal conductivity of FRFC were discussed. The results indicated that proper fiber mass fraction can play a role of refining small pores and separating large pores, improving the structural compactness, reducing the pore collapse phenomenon and optimizing the pore structure of FRFC. The three types of fibers can promote the optimization of cellular roundness and increase the proportion of pores with diameters below 400 μm. The FC with larger porosity had smaller dry density. As the fiber mass fraction increased, the thermal conductivity performed a phenomenon of first decrease and then increase. The three types of fibers with 1% mass fraction achieved relatively low thermal conductivity. Compared with the FC without fibers, the thermal conductivities of GF reinforced FC, PVAF reinforced FC and PPF reinforced FC with 1% mass fraction were decreased by 20.73%, 18.23% and 7.00%, respectively.

**Data Availability Statement:** All relevant data are within the paper.

## 1. Introduction

The conflict between the pursuit of a high quality of life and the increasing shortage of energy is becoming increasingly evident with the rapid development of the global economy [1, 2]. Building structures are developing rapidly. However, meeting the comfort, privacy and

**Funding:** This research has been supported by Suqian Sci & Tech Program (K202141); and Suqian Top 1000 Talents Training Project in 2021.

**Competing interests:** The authors declare that they have no known competing financial interests or personal relationships that could have appeared to influence the work reported in this article.

functionality of the environment by consuming large amounts of resources such as coal and electricity is contrary to the concept of green and sustainable development that has been strongly advocated [3–5]. To alleviate this problem, scholars have considered the development and application of new energy-saving insulation walls as one of the key technologies to achieve energy efficiency in building structures [6, 7].

Foamed concrete (FC) is prepared according to the matching ratio of the cementitious material slurry. The foaming agent solution is prepared into foam by specific equipment (e.g., high-pressure foaming machine), and then added to the prepared cementitious material slurry, which is a kind of lightweight porous building material after mixing, pouring and maintenance [8, 9]. FC has the characteristics of heat insulation, heat preservation, sound absorption, light weight, good fire resistance and low cost. It has become a hot spot topic in the field of building energy efficiency [10–12].

The most prominent feature of FC is high porosity, so the pore structure is an important indicator [13–15]. The pore structure is a reflection of the mesoscopic performance of FC [16]. Adjusting the mesoscopic structure of FC is an important means to change the macro performance of FC. The main pore structure parameters of FC are porosity, average pore size, pore size distribution, roundness value. However, FC suffers from low strength, cracking, high water absorption and drying shrinkage due to the presence of uniform porous structure and lack of coarse aggregates [17, 18]. In FC, the addition of three-dimensional randomly distributed fibers can change the typical brittle behavior into an elastic-plastic one, thus improving the compressive strength, tensile strength and ductility of FC. Fallianoa et al. [19] concluded that placing glass fibers directly on the bottom of the flexural members of FC can improve the flexural properties of the members. Mirza et al. [20] investigated the effect factors on FC. The results illustrated that the fiber mass fraction was 1. 0%~2. 0% can effectively control the restrained shrinkage cracking and improve the flexural toughness of FC. Daneti et al [21] found that polypropylene fibers had a great influence on improving flexural toughness and controlling the shrinkage cracking behaviors of the lightweight FC. Soleimanzadeh et al. [22] mixed polypropylene fibers and fly ash normal silicate cement-based FC together to make flexural members and investigated the flexural performance at different temperatures. The results revealed that the polypropylene fibers enhanced the flexural properties of the members and reduced the strength loss rate by 30% at 600˚C compared to the benchmark group. Falliano et al. [23] added polypropylene microfibers to FC, which made the flexural and compressive strengths of RC increase. Raj et al. [24] found that the addition of an optimum hybrid combination of synthetic and natural fibers can enhance the strength of FC and make FC more durable.

The pore structure is an important factor affecting the thermal conductivity of FC [25, 26]. When fiber is added into FC, the distribution, overlap and entanglement of fibers in the FC slurry affect the pore structure of FC. To date, there are few studies on the effects of different types of fibers on the pore structure characteristics and thermal conductivity of FC, which is crucial to the application and promotion of FC in green buildings.

With this aim, the pore structure characteristics and thermal conductivity of FC reinforced with glass fibers (GF), polyvinyl alcohol fibers (PVAF) and polypropylene fibers (PPF) were prepared and tested in this article. The adhesion of with different mass fractions of GF, PVAF and FFF to cementitious base was investigated by the SEM images of fibers reinforced FC (FRFC). The pore structure characteristic parameters of FRFC were analyzed using Photoshop software and Image Pro Plus (IPP) software. The effects of different mass fractions and lengths of three types of fibers on the thermal conductivity of FRFC were discussed.

## 2. Preparation and performance test of FC reinforced fiber specimens

### 2.1 Preparation of FC reinforced fiber specimens

In this article, the effect of fibers on the pore structure and thermal conductivity of foam concrete were investigated. In addition, the overall compressive strength of foam concrete is low, roughly in the range of 0.5 ~ 10MPa, and it has been found that the appropriate amount and type of fibers can improve the pore structure and improve the mechanical properties of foam concrete. Therefore, the foam concrete reinforced with fibers prepared in this article not only meets the basic mechanical properties but also has high thermal conductivity.

The raw materials used for the test specimens of FRFC were mainly P • O 42.5 ordinary Portland cement, I low calcium fly ash, composite plant protein high-efficiency foaming agent (dilution ratio was 30 times), 540P polycarboxylic acid water reducer and fibers. The types of fibers were GF (see Fig 1(A), PVAF (see Fig 1(B)) and PPF (see Fig 1(C). The basic properties of the above three types of fibers are listed in Table 1.

Firstly, a certain amount of cement, fly ash and water reducer were poured into the mixing bucket and stirred with a dry spoon; water was poured after mixing evenly; they were stirred with a hand-held high-speed mixer and then stopped quietly to obtain the slurry I. Then, fibers were gradually added to the slurry I and stirred at low speed to prevent lumping. After the fibers were completely added, they were stirred again at high speed to mix evenly to obtain slurry II. Meanwhile, the foaming agent and water were stirred at high speed to get foam. An appropriate amount of foam was taken and stirred together with slurry II to obtain homogeneous slurry III. Finally, slurry III was poured into a test mold. The surface of FRFC in the test mold was scraped flat. It was covered with plastic wrap and left at (20 ± 2)°C ambient temperature for 24 hours. The mold was removed to obtain the test specimens. Fig 2 gives the preparation process of FCRC. Table 2 lists the mix ratios of FRFC reinforced with three types of fibers.

### 2.2 Performance tests of FRFC

To investigate the pore structure characteristics and thermal conductivity of FRFC, the following four types of tests were carried out on FRFC.

(1) SEM tests.   To further investigate the bonding of the fibers to the FC matrix, the specimens were dried and gold sprayed. Then, its micromorphology was characterized using a FEI Scios 2 HiVac scanning electron microscope.

(2) Dry density tests.   Three FRFC test blocks of 10 mm ×100 mm ×100 mm were prepared for each label in Table 1. The test blocks were dried continuously in an oven at (60 ± 5)°C until the mass difference was less than or equal to 1g twice within 4 h. After cooling to room

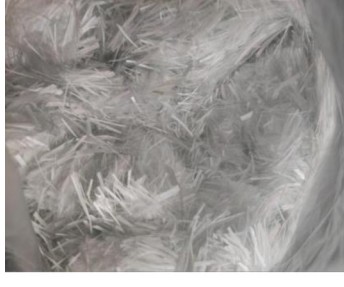 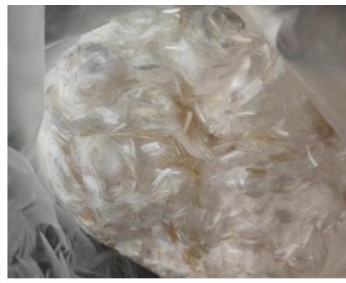 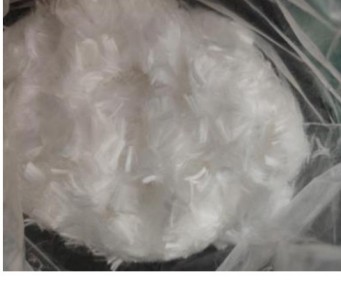

|  (a) GF | (b) PVAF | (c) PPF |

**Fig 1. Photos of three types of fibers.** (a) GF, (b) PVAF, (c) PPF.

**Table 1. Basic properties of the three types of fibers.**

| Type | Ultimate tensile strength (MPa) | Modulus of elasticity (GPa) | Ultimate elongation (%) | Equivalent diameter ($10^2$ μm) | Length (mm) | Density (g/cm$^3$) |
|------|--------------------------------|-----------------------------|-------------------------|--------------------------------|-------------|--------------------|
| GF   | >2600                          | >80                         | 3~4                     | 13                             | 6~15        | 2.63               |
| PVAF | 1400~1600                      | 35~39                       | 6~8                     | 15                             | 6~15        | 1.50               |
| PPF  | >486                           | >4.8                        | >18                     | 20                             | 6~15        | 0.91               |

temperature, their masses were weighed using an electronic scale with an accuracy of 0.01g. The test was performed according to the FC standards in [27]. The dry density of the test block was calculated using Formula (1). The arithmetic average of dry density of the three test blocks in each label was taken as the dry density $\rho_0$ of FRFC test blocks in each label.

$$\rho_0 = \frac{m_0}{V} \times 10^6 \tag{1}$$

Where: $m_0$ and $V$ were absolute dry mass and absolute volume respectively.

(3) Porosity tests.   FRFC specimens were cut into cubes to measure the absolute density using the drainage method. The true density of the specimen was tested with reference to the standards GB/T208-2014 [28]. The porosity $P$ was calculated using Formula (2).

$$P = \left(1 - {}^{\rho_0}/_{\rho}\right) \times 100\% \tag{2}$$

In Formula (2), $P$ and $\rho$ were porosity and true density respectively.

(4) Thermal conductivity tests.   The thermal conductivity of FC reflects its thermal insulation performance, the lower the thermal conductivity, the better the thermal insulation performance [29]. The size of the test blocks is 300 mm × 300 mm × 30 mm, and it is demolded and cured under standard conditions. The test blocks were dried in a drying oven at (105 ± 5)°C for 24 hours and then cooled to room temperature. They were tested for thermal conductivity using a JTRG-Ⅲ thermal conductivity tester according to the standards GB 10294–2008. Each test block was tested 3 times, and their average value was taken as the thermal conductivity.

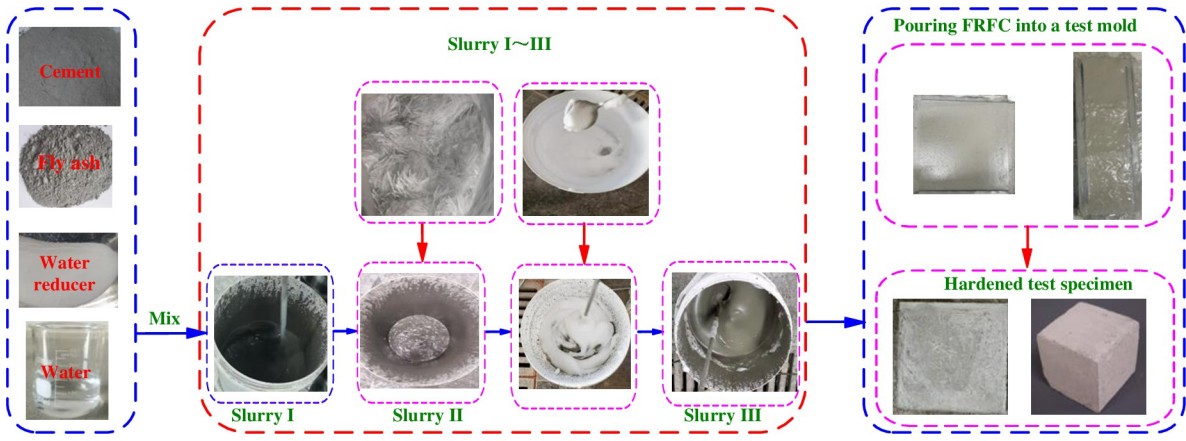

**Fig 2. Preparation process of FRFC.**

**Table 2. Mix ratios of FRFC.**

| Label | Cement/Fly ash | W/B | Water reducer (%) | Types of fibers | $m_f$ (%) | $V_s$:$V_f$ |
|-------|----------------|-----|-------------------|-----------------|-----------|-------------|
| 0 | 0.7/0.3 | 0.4 | 0.1 | / | 0 | 1:0.87 |
| A1 | 0.7/0.3 | 0.4 | 0.1 | GF | 1 | 1:0.87 |
| A1.5 | 0.7/0.3 | 0.4 | 0.1 | GF | 1.5 | 1:0.87 |
| A2 | 0.7/0.3 | 0.4 | 0.1 | GF | 2 | 1:0.87 |
| B1 | 0.7/0.3 | 0.4 | 0.1 | PVAF | 1 | 1:0.87 |
| B1.5 | 0.7/0.3 | 0.4 | 0.1 | PVAF | 1.5 | 1:0.87 |
| B2 | 0.7/0.3 | 0.4 | 0.1 | PVA | 2 | 1:0.87 |
| C1 | 0.7/0.3 | 0.4 | 0.1 | PPF | 1 | 1:0.87 |
| C1.5 | 0.7/0.3 | 0.4 | 0.1 | PPF | 1.5 | 1:0.87 |
| C2 | 0.7/0.3 | 0.4 | 0.1 | PPF | 2 | 1:0.87 |

Note: $m_f$ is the mass fraction of fibers, which is equal to fiber mass divided by the sum of cement mass and fly ash mass; $V_s$ is the volume of the slurry; and $V_f$ is the volume of the foam.

## 2.3 Processing methods for binarized images

The cross sections of the 10 labels of FRFC specimens in Table 1 were tested. One specimen from each lable was selected and cut from the casting surface to the bottom surface with a small hacksaw along the direction perpendicular to the forming surface. The surface was sanded with 240 grit sandpaper until the surface was smooth and flat. The powder on the surface and inside the hole of the specimen was blown off with an air gun. The test specimen with the attached scale was photographed with a SONY α7rIII camera. Fig 3 gives the cross-section

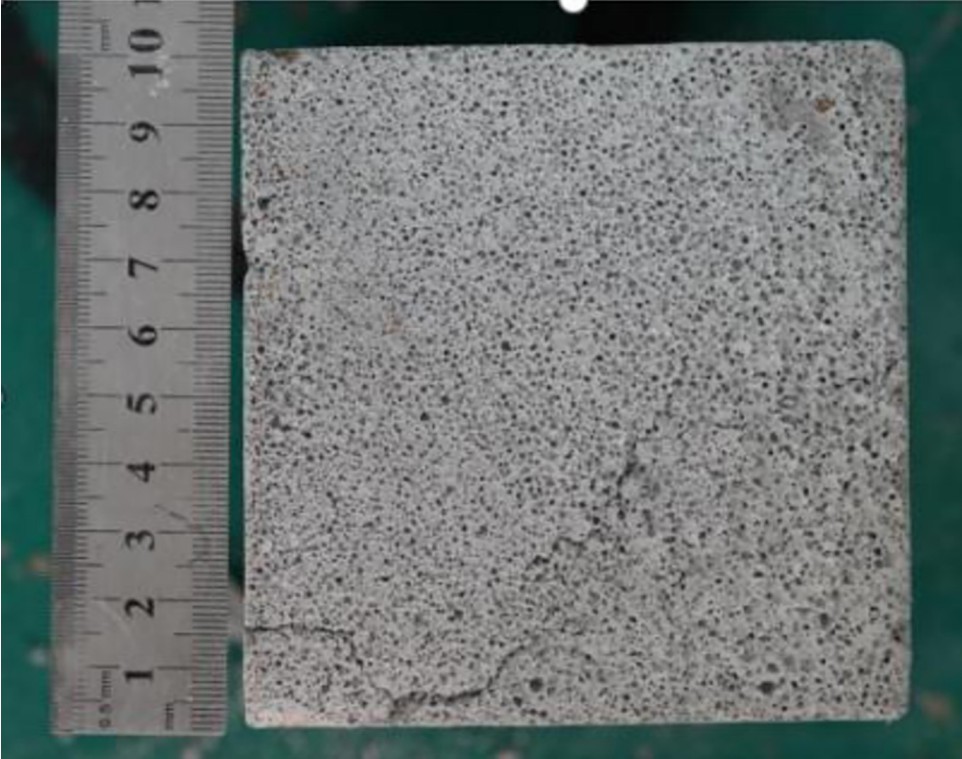

**Fig 3. Cross-section image of FRFC specimen.**

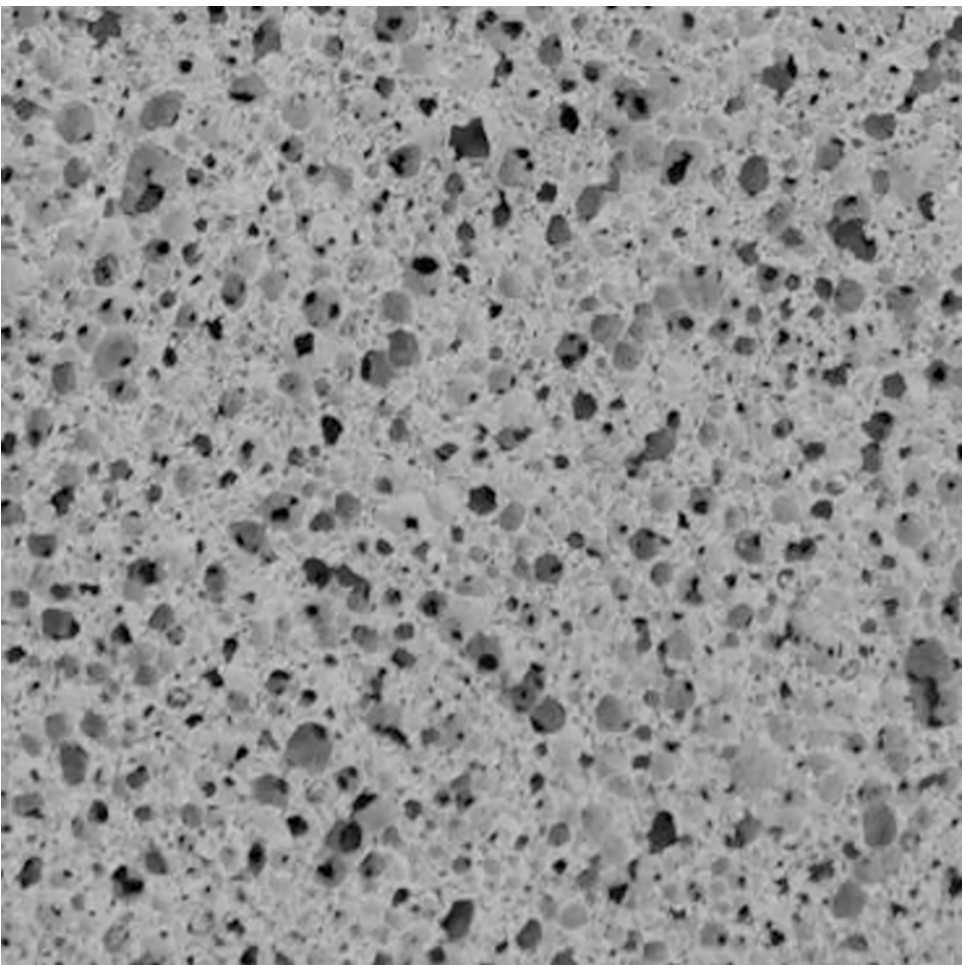

**Fig 4. Selected cross-section image.**

image of FRFC specimen. The cross-section size of the specimen was 100 mm × 100 mm. The scale on Fig 3 was 0.5.

To ensure a reasonable selection of images, four images were randomly captured in each section according to the scale in the image using Photoshop software. The image without defects was selected as object, as shown in Fig 4. The size of each image was 20 mm × 20 mm. The scale on Fig 4 was 3. The cropped images were continued to be de-grayed using Photoshop software, and then the degree of binarization was adjusted by adjusting the threshold.

There were two ways to select holes in the binarized images, namely manual selection and automatic recognition. With hundreds or even thousands of holes in a cross-section, the Image Pro Plus (IPP) software was chosen for this experiment to automatically identify the holes. Statistics on pore size distribution, porosity and shape factor were obtained using the IPP software.

## 3 Results and discussion

### 3.1 Micromorphological analysis

Fig 5 shows the SEM images of the adhesion of GF with 0%, 1%, 1.5% and 2% mass fractions to cementitious base. The resolution of the SEM images was. The scale of the SEM images was

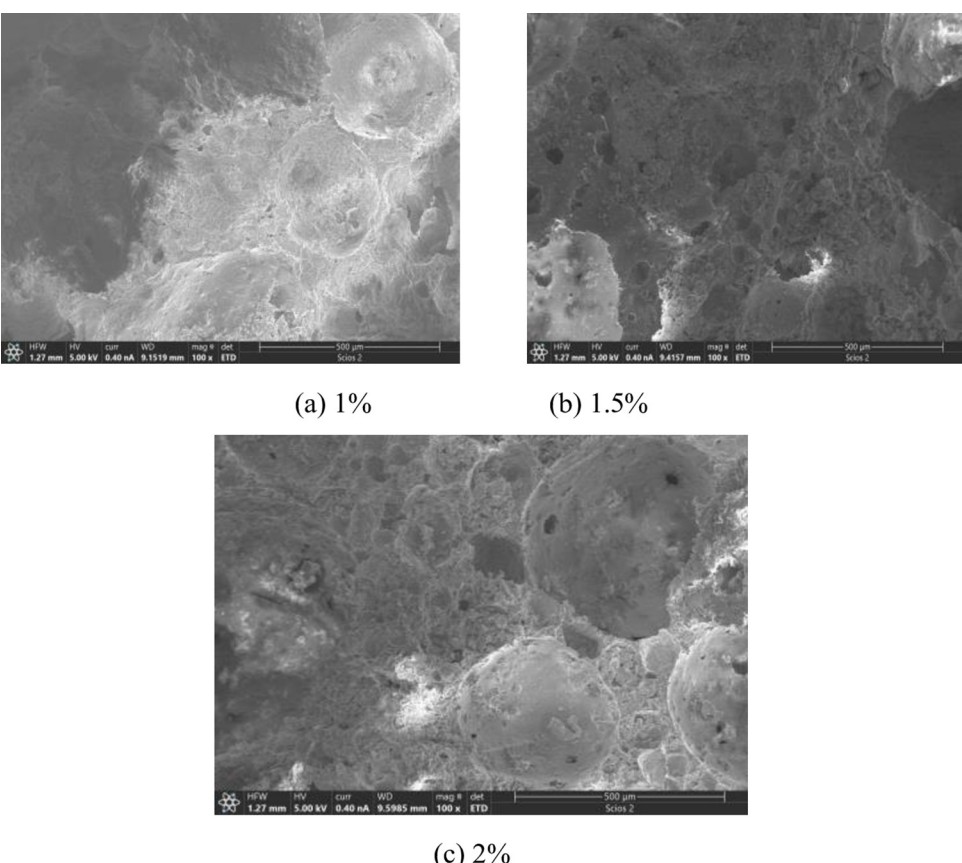

(a) 1%          (b) 1.5%

(c) 2%

**Fig 5. Adhesion of different mass fractions of GF to cementitious base.** (a) 1%, (b) 1.5%, (c) 2%.

100. When the mass fractions of GF are 1%, 1.5% and 2%, the fibers can be uniformly dispersed in the cement hydration products and no obvious clumping phenomenon was observed. This was because the elastic modulus of GF was larger than that of PPF or PVAF, the fibers had a large stiffness under the same length condition, the clumping phenomenon was not obvious when the mass fraction was small.

Fig 6 shows the SEM images of the adhesion of PVAF with 0%, 1%, 1.5% and 2% mass fractions to cementitious base. The resolution of the SEM images was. The scale of the SEM images was 100. The surface of PVAF was rough and uneven. As shown in Fig 6(A) and 6(B), when the mass fraction of PVAF was small, it can snap tightly onto the cementitious material and improve the toughness and compaction of the cementitious base. As shown in Fig 6(C), when the mass fraction of PVAF was 2%, the excess fibers cannot be uniformly dispersed in the cementitious base, and the bending and winding of fibers appeared, which were randomly gathered together and randomly distributed on the surface and inside of the FC. The above microscopic results were consistent with the macroscopic phenomenon.

Fig 7 shows the SEM images of the adhesion of PPF with 0%, 1%, 1.5% and 2% mass fractions to cementitious base. The resolution of the SEM images was. The scale of the SEM images was 100. When the mass fraction of was 1%, there was good wrapping of the cementitious base. There were almost no microcracks at the interface between the two. No significant loosening of the fibers was observed (see Fig 7(A)), which was helpful to mitigate the occurrence of microcracks in the cement slurry after hardening. As the mass fraction of PPF was increased

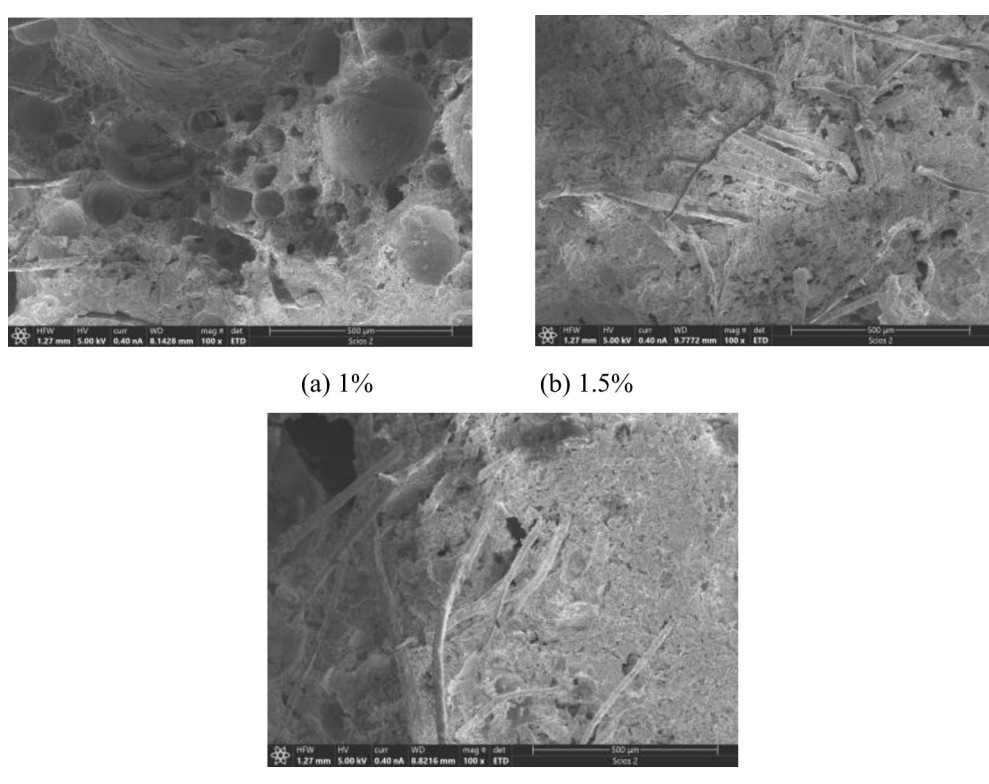

(a) 1%          (b) 1.5%

(c) 2%

**Fig 6. Adhesion of different mass fractions of PVAF to cementitious base.** (a) 1%, b) 1.5%, (c) 2%.

to 2%, a large number of fibers were disordered due to its own elastic modulus was low and randomly pulled out due to the shrinkage stress of the anisotropic cementitious base (see Fig 7(C)).

From above analysis, it can be concluded that the appropriate mass fraction of the fibers can improve the structural compactness of FRFC. It reduced the pore collapse phenomenon, optimized the pore structure and improve the integrity of FRFC.

### 3.2 Pore structure properties

**3.2.1 Binarized images of the pore structure.** The bubbles in FRFC were in a thermodynamically unstable state. As the slurry solidified and hardened, the pores gradually stopped splitting and deforming, and their size and shape tend to stabilize. The binarized images of FRFC with different fiber types and mass fractions are shown in Figs 8–10. Compared with the FC without fibers, the pore size of the air hole in FRFC with 1% mass fraction of GF, PVAF or PPF was decreased. In FRFC, the shape of the air pores was relatively round and the distribution of it was uniform; the pore size distribution was wider and number of obviously macropores was significantly reduced. It indicated that fibers limited the cracking and growth of macropores. Therefore, the pore size was decreased overall, and the pore structure of the FC was optimized. When the fiber mass fraction was 2%, the workability during stirring of FRFC was poor, indicating that the fibers were excessive. In particular, after the PVA fibers were mixed in the FC, the agglomeration phenomenon of the hardened FC section was more obvious. At the same time, small pores were connected to form macropores, and some of the pores collapsed, indicating that excessive fibers affected the uniformity of the pore size.

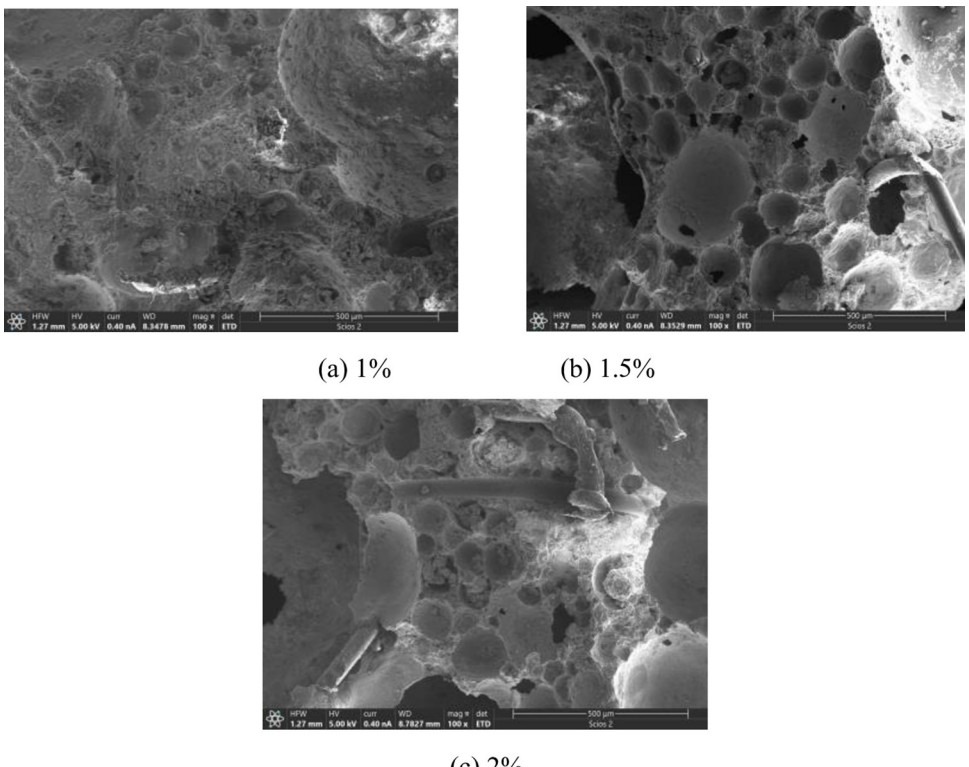

(a) 1%          (b) 1.5%

(c) 2%

**Fig 7. Adhesion of different mass fractions of PPF to cementitious base.** (a) 1%, (b) 1.5%, (c) 2%.

**3.2.2 Pore size.** Fig 11 gives the histograms of pore size distribution of FC reinforced with different types of fibers. For one type of FRFC in Fig 11, the proportion is between the number of pores with the same size and the total number of all pores with different sizes. The mass fraction s of the fibers in one type of FRFC were 0%, 1%, 1.5%, 2%, respectively. Compared with the FC without fibers, the proportion of the small pores with diameters below 400 μm increased significantly as the fiber mass fraction increased, and the proportion of large pores with diameters above 400 μm showed a decreasing trend, indicating that the appropriate fiber

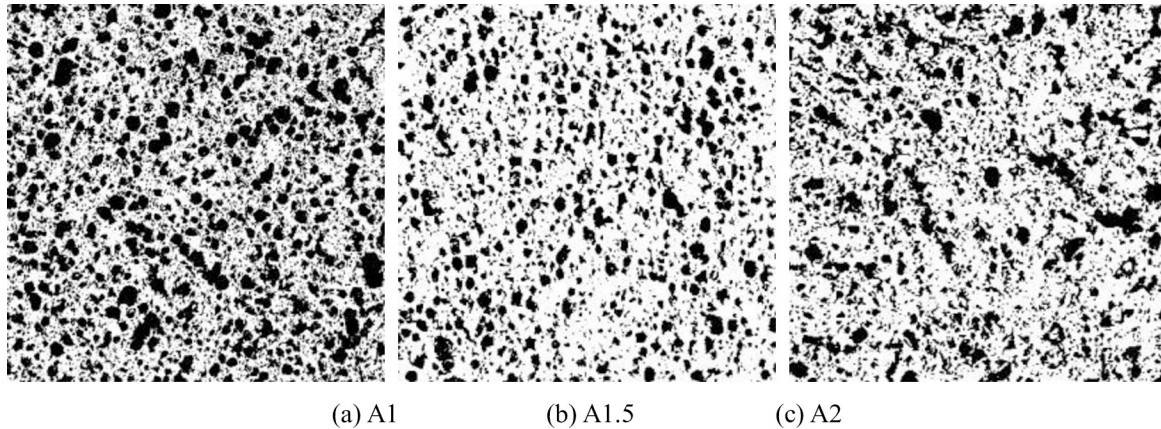

(a) A1                    (b) A1.5                    (c) A2

**Fig 8. Binarized images of FC reinforced with GF.** (a) A1, (b) A1.5, (c) A2.

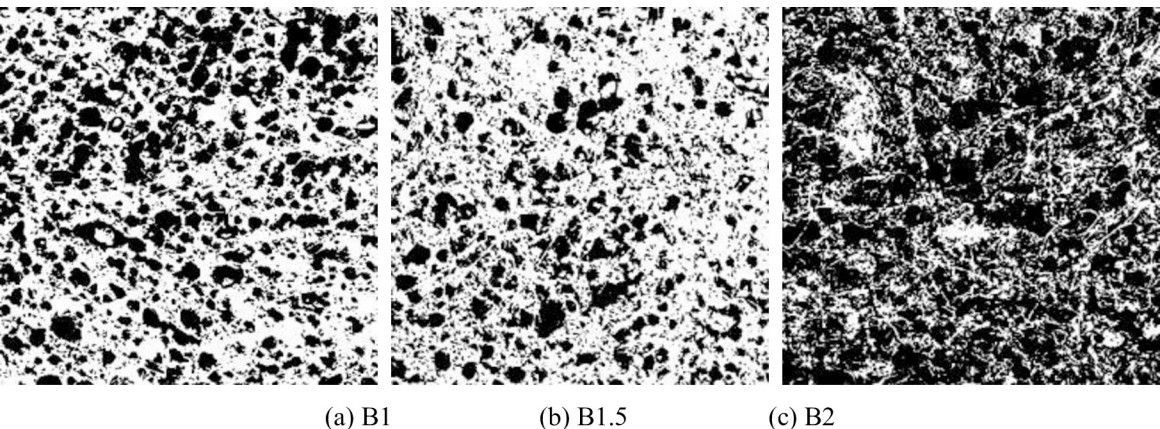

(a) B1 (b) B1.5 (c) B2

**Fig 9. Binarized images of FC reinforced with PVAF.** (a) B1, (b) B1.5, (c) B2.

mass fraction effectively alleviated the phenomenon of water secretion and segregation and improved the compactness of the hardened FC. When the mass fraction of PPF was 1%, the proportion of small pores with diameters below 200 μm reached 51%, which had a great effect on the optimization of pore size. When the mass fraction of GF was 1%, the proportion of small pores with diameters below 200 μm was 42%. When the mass fraction of PVAF was 2%, the proportion of small pores with diameters below 200 μm was 34%. When the pore size was below 600 μm (relatively small), the proportion of FC reinforced with PVAF was high, which contains the pore size range of 200–400 μm. However, with the gradual increase in pore size, the proportion of the FC without fibers gradually increased, exceeding that of foam concrete with PVAF. This was mainly due to the fact that the addition of fibers can form a stable structure of three-dimensional network lap, which played the role of support and division and helped to form smaller and uniform size pores. It can be concluded that the FC reinforced with 1% mass fraction of PPF mostly had uniform and small closed pores inside, and the appropriate amount of fibers played the role of filling and bridging, which is helpful to improve the thermal conductivity and increases the mechanical strength.

**3.2.3 Pore shape.** The shape characteristics of air pores can be described by the value of shape factor S, which is mainly used to evaluate the degree of deviation of the pores from

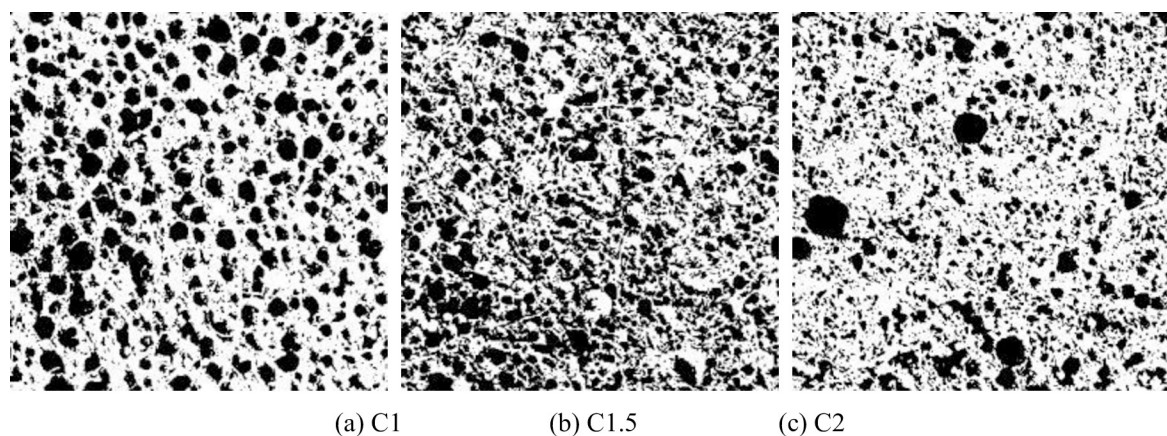

(a) C1 (b) C1.5 (c) C2

**Fig 10. Binarized images of FC reinforced with PPF.** (a) C1, (b) C1.5, (c) C2.

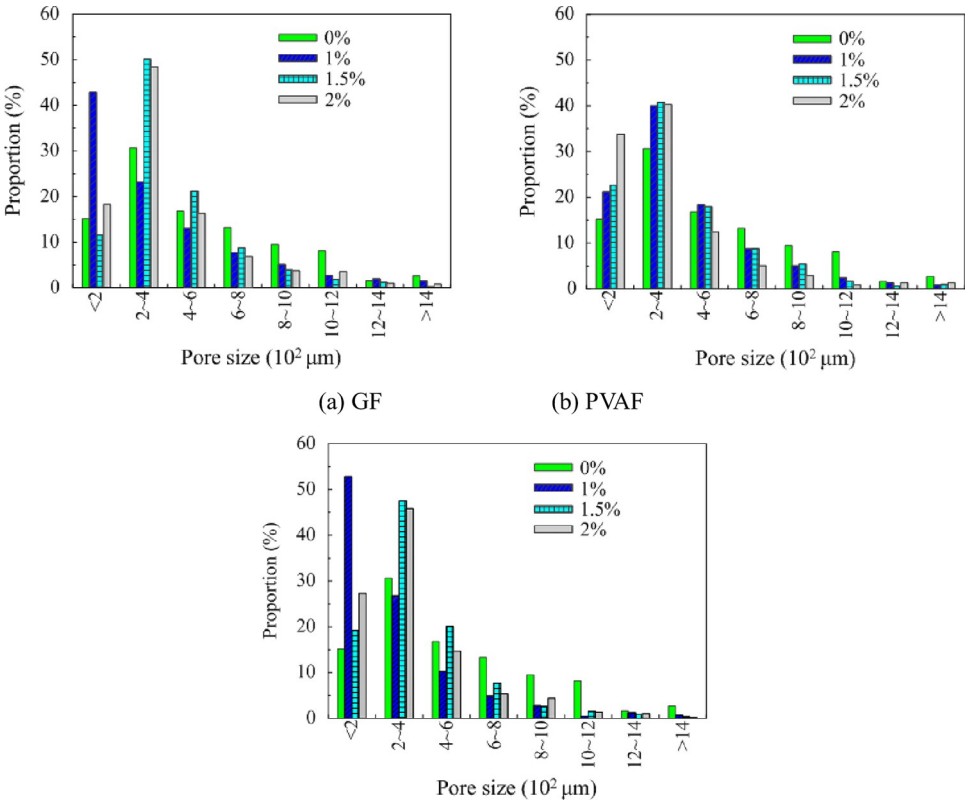

**Fig 11. Histograms of pore size distribution of FRFC.** (a) GF, (b) PVAF, (c) PPF.

spherical. The closer the pore shape is to a regular sphere, the value of $S$ is close to 1 [30]. The histograms of the pore shape factor distribution of FRFC are shown in Fig 12. For one type of FRFC in Fig 12, the proportion is between the number of pore shape factor with the same value and the total number of all pores with different shape factors. The mass fraction s of the fibers in one type of FRFC were 0%, 1%, 1.5%, 2%, respectively. Since the FC was mixed with fly ash in this test, an appropriate amount of fly ash had a micro-aggregate effect, which optimized the particle gradation and improved the pore shape. Compared with the FC without fibers, the proportion of $S$ in the range of 1 to 1.2 increased significantly for the FC reinforced with fibers, while the proportion of $S \geq 1.2$ decreased for the FC reinforced with fibers. When the mass fraction of GF was 1%, 1.5% and 2%, respectively, the proportions of $S$ in the range of 1~1.2 increased by 29.8%, 32.5%, 20%. When the mass fractions of PVAF were 1%, 1.5% and 2%, respectively, the proportions of $S$ in the range of 1~1.2 increased by 30%, 27.5%, 22.5%. When the mass fractions of PPF were 1%, 1.5% and 2%, respectively, the proportions of $S$ in the range of 1~1.2 increased by and 57.5%, 45%, 15%. It indicated that these three types of fibers played a role in optimizing the roundness value and reducing the occurrence of connected pores. Among them, PPF with 1% mass fraction had the most significant effect on improving the pore shape.

**3.2.4 Porosity.** FC is a gas-solid two-phase mixture, the heat transfer between the gas and the solid is achieved by the collision of gas molecules and thermal vibration of atoms (molecules) respectively. Therefore, its heat transfer is the result of the combined effect of the two aforementioned heat transfer methods, while the small amount of heat transfer through thermal radiation is negligible. In FRFC, the larger the proportion of cementitious base, the larger

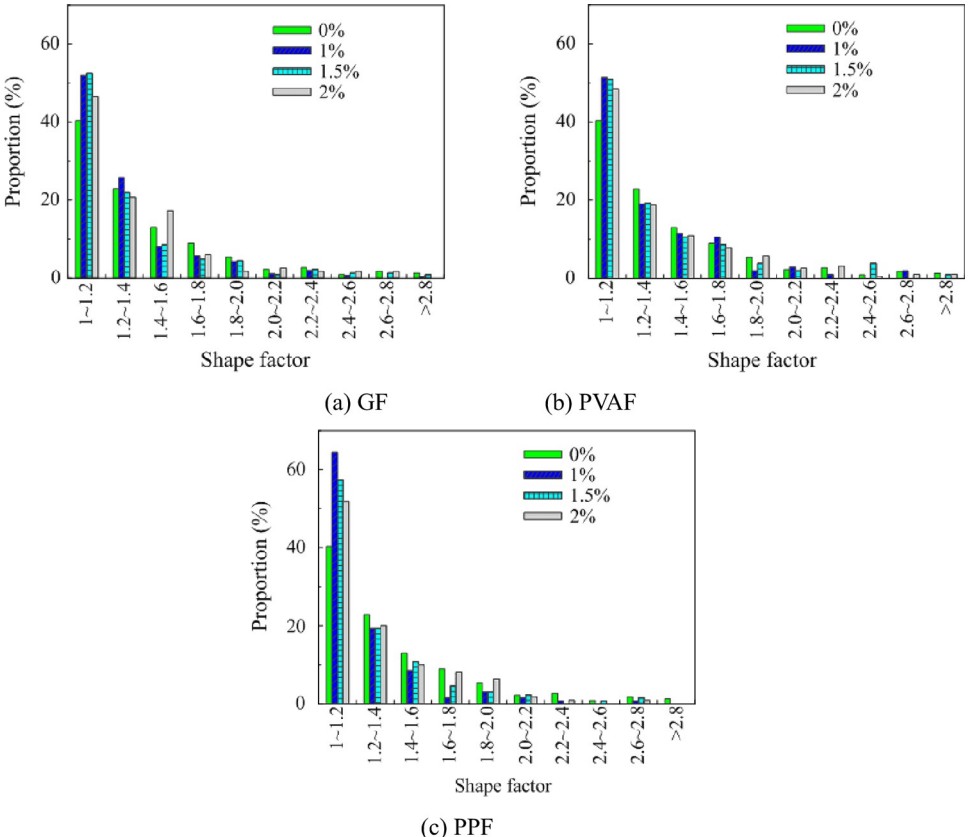

**Fig 12. Histograms of pore shape factor distribution of FC reinforced with different fibers.** (a) GF, (b) PVAF, (c) PPF.

the proportion of heat transfer through heat conduction, which increased the thermal conductivity. As the porosity increased, the smaller the proportion of solid phase in the FC, the dry density gradually decreased. The relationship between dry density ($\rho_0$) and porosity ($P$) of FRFC was fitted, as shown in Fig 13. The fitting curve can be obtained in Formula (3). The fitting correlation coefficient was 0.9795, indicating that the relationship between dry density and porosity approximated to a unitary linear equation and the correlation between them was good.

$$\rho_0 = -16.08P + 1449.42 \tag{3}$$

### 3.3 Effect of fibers on the thermal conductivity of FRFC

**3.3.1 Types and mass fractions of fibers.** The effect of the mass fraction on the thermal conductivity of FRFC is shown in Fig 14. Heat was conducted through vibration, the average pore size of air pores gradually decreased after the fibers were added, and the thermal conductivity of air was smaller than that of solid, so the thermal insulation performance of FC was improved. Therefore, the thermal conductivity performed a downward trend as the fiber mass fraction increased. When the FC was mixed with 1% fiber mass fraction, the thermal conductivities of the FC reinforced with GF, PVAF and PPF were decreased by 20.73%, 18.23% and 7.00%, respectively. It indicated that an appropriate fiber mass fraction pile up with each other

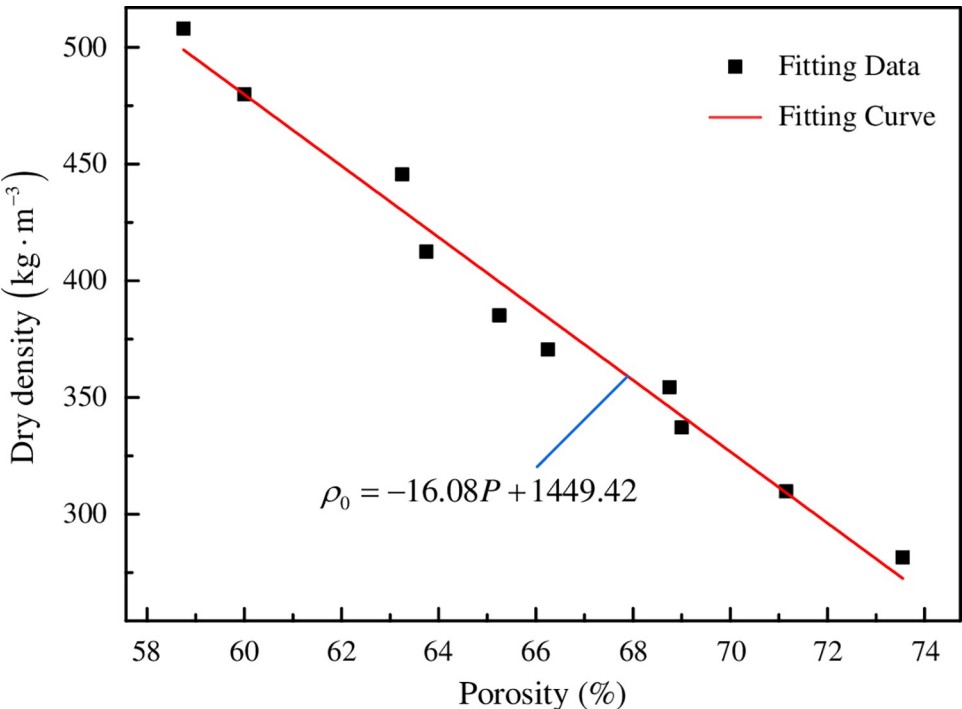

**Fig 13. The relationship between dry density and porosity of FRFC.**

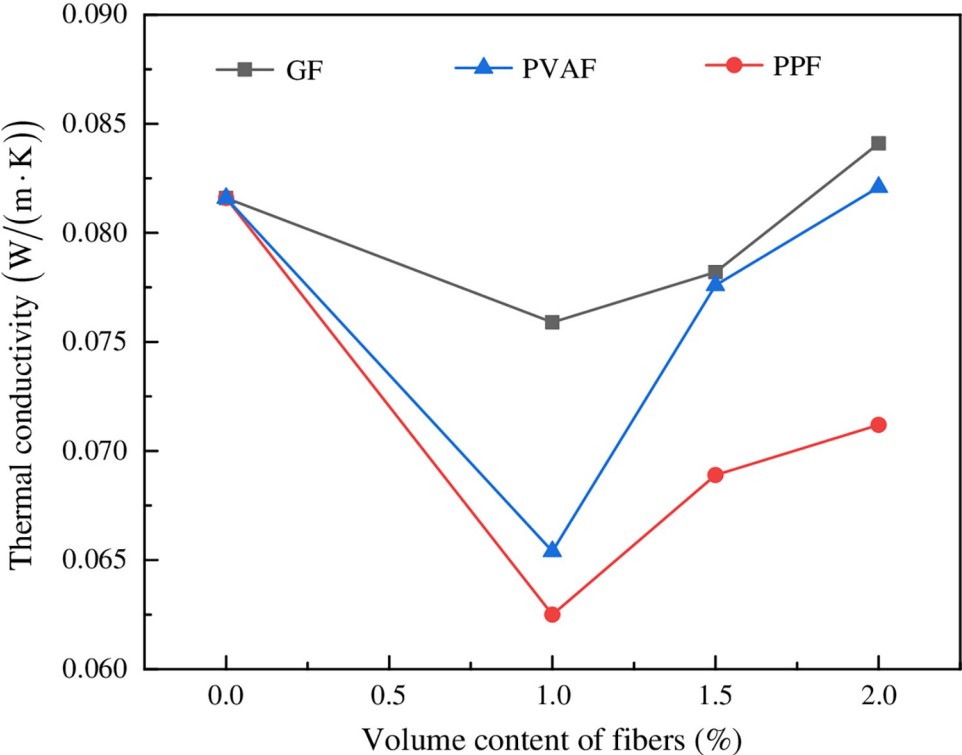

**Fig 14. Effect of mass fraction of fibers on the thermal conductivity of FRFC.**

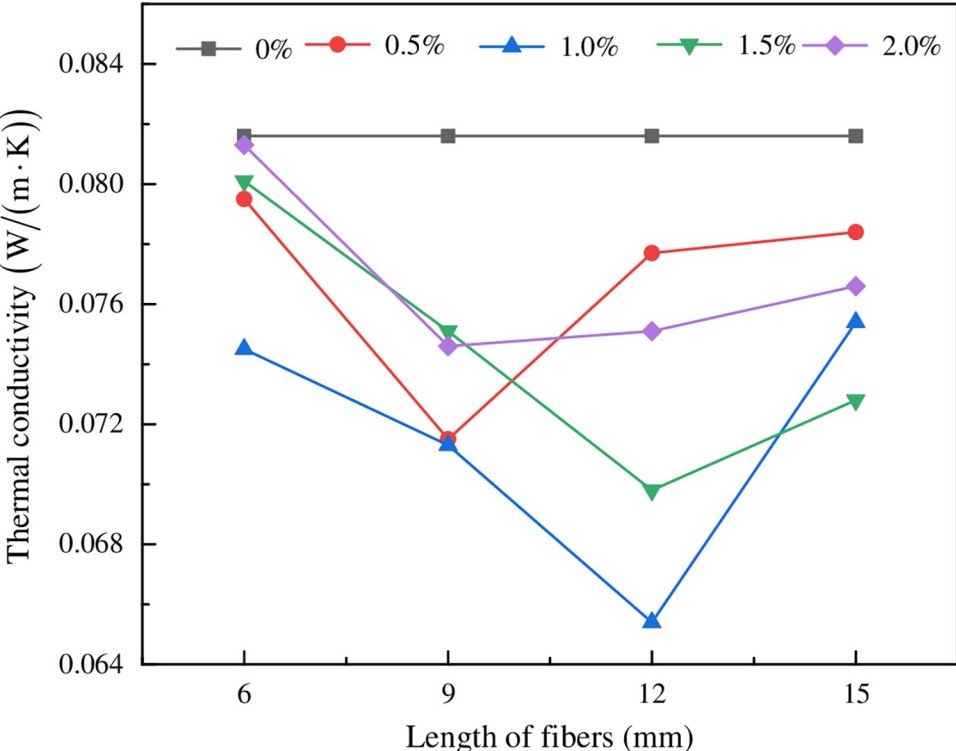

**Fig 15. Effect of lengths of fibers with different mass fraction s on thermal conductivity of FRFC.**

to form a network structure, and the large pores were divided into small pores by the fibers, which was conducive to improving the stability and uniformity of the pore structure. Since the thermal conductivity of the porous phase was lower than that of the solid phase, it increased the heat transfer resistance and thus reduced the heat transfer rate, resulting in a decreasing trend of thermal conductivity. When the fiber content gradually increased, the fibers entangled and agglomerated with each other, reducing the fluidity of the slurry, while the excess fibers caused part of the foam to be destroyed in the mixing process, leading to a decrease in the porosity of the hardened FC, which showed a tendency of increasing thermal conductivity.

**3.3.2 Lengths of fibers.** Four groups of PPF with different lengths of 6 mm, 9 mm, 12 mm and 15 mm were added to the FC. The effect of fiber lengths with different mass fractions on the thermal conductivity of FRFC was investigated in Fig 15. Compared with the FC without fibers, the thermal conductivity of FC reinforced with different lengths of PPF was significantly decreased, and the thermal conductivity performed first decreased and then increased as the length of PPF increased. Among them, when the PPF content was 0.5%, 1.0%, 1.5% and 2%, the minimum thermal conductivities of the FC reinforced with 0.5%, 1.0%, 1.5% and 2% mass fractions of PPF were 0.072 W/(m•K), 0.065 W/(m•K), 0.070 W/(m•K) and 0.075 W/(m•K), respectively. Compared with those of the FC without fibers, they were decreased by 12.1%, 20.7%, 14.6% and 8.5%, respectively.

## 3.4 Discussion

Fibers can effectively increase the proportion of small pores with diameters below 600 μm in the FC. The proportion of the shape factor in the range of 1 to 1.2 of the FC without fiber was 40%. After adding three types of fibers to the FC, the proportion of the shape factor in the

range of 1 to 1.2 were greater than 40%. The changes in pore size and shape indicated that the fibers can effectively partition the air bubbles, making the pores smaller, closer in size, and more uniformly distributed, thus optimizing the roundness value and increasing the proportion of shape factor close to 1. Compared with the FC without fiber, the proportion of small pores with the diameters below 200 μm in FC can be increased by adding different types of fibers with different mass fractions. Fibers can inhibit the formation of a large number of large pores and was conducive to the formation of a uniform size pore structure, thus ensuring stable thermal insulation performance, which was consistent with the experimental test result that the appropriate mass friction of fibers can reduce the thermal conductivity obtained in this article (Compared with the FC without fibers, the thermal conductivity of the FC with the 1% mass friction of PPF, PVAF and GF was decreased by 20.73%, 18.23% and 7.00%, respectively).

## 4 Conclusions

To investigate the effect of three types of fibers with different mass fractions on the pore structure characteristics and thermal conductivity of FRFC, SEM tests, dry density tests, porosity tests, and thermal conductivity tests were carried out. The main conclusions can be drawn as following:

1. Appropriate fiber mass fraction can play a role of refining small pores and separating large pores, improve the structural compactness, reduce the pore collapse phenomenon and optimize the pore structure of FRFC. When the fiber mass fraction of fibers increased to 2%, the agglomeration induced the macropores were connected and the air pores collapsed in the hardened FC.

2. Adding an appropriate amount of fly ash to FC can optimize the particle gradation and effect, and adding fibers can improve the pore shape factor. 1% mass fraction of PPF was the most significant for improving the pore shape.

3. Based on the analysis results of the pore structure characteristics using the IPP software, the smaller the proportion of solid phase in the FC, the lower the dry density. The fitting formula between dry density ($\rho_0$) and porosity ($P$) of FRFC was $\rho_0 = -16.08P + 1449.42$.

4. The thermal conductivity of FC can be decreased by adding GF, PVAF and PPF. As the fiber mass fraction increased, the thermal conductivity performed a phenomenon of first decrease and then increase. The three types of fibers with 1% mass fraction achieved relatively low thermal conductivity. Compared with the FC without fibers, the thermal conductivities of GF, PVAF and PPF with 1% mass fraction were decreased by 20.73%, 18.23% and 7.00%, respectively. PPF and PVAF improved the thermal conductivity of the FC better than GF.

## Author Contributions

**Conceptualization:** Lin Li.

**Data curation:** Lin Li, Wei Wang, Yu Wang.

**Formal analysis:** Lin Li.

**Funding acquisition:** Lin Li.

**Investigation:** Lin Li, Wei Wang.

**Methodology:** Yu Wang.

**Project administration:** Lin Li, Mei-Ling Zhuang.

**Resources:** Lin Li, Yu Wang.

**Software:** Wei Wang, Dongxu Li.

**Supervision:** Mei-Ling Zhuang.

**Validation:** Yu Wang.

**Visualization:** Yu Wang, Dongxu Li.

**Writing – original draft:** Lin Li.

**Writing – review & editing:** Mei-Ling Zhuang.

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
