## [Decision Letter · Decision Letter 0]

12 Dec 2022

PONE-D-22-30437Experimental study on Pore Structure Characteristics and Thermal Conductivity of Fibers Reinforced Foamed ConcretePLOS ONE

Dear Dr. Zhuang,

Thank you for submitting your manuscript to PLOS ONE. After careful consideration, we feel that it has merit but does not fully meet PLOS ONE’s publication criteria as it currently stands. Therefore, we invite you to submit a revised version of the manuscript that addresses the points raised during the review process.

Please go through the comments of reviewers and make the recommendations accordingly.

We look forward to receiving your revised manuscript.

Kind regards,

Yasir Nawab, PhD

Academic Editor

PLOS ONE

Journal Requirements:

"This research has been supported by National Natural Science Foundation of China (51872137); Suqian

Science and Technology Program (K202141); and Suqian Top 1000 Talents Training Project in 2021."

"This research has been supported by National Natural Science Foundation of China (51872137); Suqian Science and Technology Program (K202141); and Suqian Top 1000 Talents Training Project in 2021."

Reviewers' comments:

Reviewer's Responses to Questions

**Comments to the Author**

1. Is the manuscript technically sound, and do the data support the conclusions?

Reviewer #1: No

Reviewer #2: Yes

2. Has the statistical analysis been performed appropriately and rigorously? 

Reviewer #1: Yes

Reviewer #2: No

3. Have the authors made all data underlying the findings in their manuscript fully available?

Reviewer #1: Yes

Reviewer #2: No

4. Is the manuscript presented in an intelligible fashion and written in standard English?

Reviewer #1: No

Reviewer #2: Yes

5. Review Comments to the Author

Reviewer #1: Authors in this manuscript prepared the various concrete samples to evaluate their pore structure characteristics and thermal conductivity. But the manuscript needs to be improved for publishing in the journal.

1. The Abstract is so ambiguously written, and sentences are too long to understand. It was very hard for the reviewer to understand the combination of fibers and concrete just my reading the Abstract. Please rearrange the sentence structure along with clear explanation.

2. Most important point is that you are considering the construction material but how the fibers concentration and their length are affecting mechanical properties? This should be the main concern first. Let’s suppose in your opinion the 1% has better results in thermal insulation but 2% has moderate results with improved mechanical performance. Please explain about this?

3. Below Fig. 6, after the Fig. 7 explanation author has directly mentioned see Fig. 13 (a). It should be then placed after Fig. 7 as Fig. 8.

4. Many term are unclear like in Figure 11, what does proportion mean? The author needs to explain this for better understanding.

5. First author needs to explain the relation of proportion and pore size relation then address comment 6.

6. The results are simply explained which the reader can understand by seeing the results. Author need to explain with reason what phenomenon tend the proportion increase with PVAF when moving towards 2-4 micrometer.

7. Authors need to add the discussion also in Result and Discussion section. Reviewer rarely can find any discussion to support the results.

8. In SEM images, different images have different resolutions. But when we are comparing the same factor in different combinations, we need to keep the resolution same. Also, you need to write the resolution separately as it is blur in images.

9. Whole English of the manuscript needs to be revised with the help of professional reviewer.

Reviewer #2: The article titled “Experimental study on Pore Structure Characteristics and Thermal Conductivity of Fibers Reinforced Foamed Concrete” deals with an experimental investigation for effect of different fibers on the pore structure characteristics and thermal conductivity of foamed concrete (FC). Samples were manufactured by adding, glass fibers (GF), polyvinyl alcohol fibers (PVAF) and polypropylene fibers (PPF) with different volume contents (0%，1%，1.5% and 2%) to the Portland cement, fly ash and plant protein foaming agent.. The research is interesting, and methodology used is relevant. A clear materials and methods section are missing. Discussions lacks rigorousness and novelty is not clearly stated. Therefore, in current form, the article cannot be recommended for publication. It is recommended to accept the article subjected to following minor revisions.

1. Literature review is good but citation of articles on concrete made with different type of fibers as well as with different techniques are not many. Authors need to provide updated state of the art. They need to add some article reporting mechanical behavior of different fiber/fabric reinforced concretes with different techniques: for example:

Muhammad Imran Khan, Muhammad Umair, Khubab Shaker, Abdul Basit, Yasir Nawab & Muhammad Kashif (2020) Impact of waste fibers on the mechanical performance of concrete composites, The Journal of The Textile Institute, 111:11, 1632-1640,

DOI: 10.1080/00405000.2020.1736423

Umair, M., Khan, M.I., Nawab, Y. (2020). Green Fiber-Reinforced Concrete Composites. In: Kharissova, O., Martínez, L., Kharisov, B. (eds) Handbook of Nanomaterials and Nanocomposites for Energy and Environmental Applications. Springer, Cham. https://doi.org/10.1007/978-3-030-11155-7_113-1

Ali M, Khan MI, Masood F, Alsulami BT, Bouallegue B, Nawaz R, Fediuk R. Central composite design application in the optimization of the effect of waste foundry sand on concrete properties using RSM. InStructures 2022 Dec 1 (Vol. 46, pp. 1581-1594). Elsevier.

2. Authors need to add one paragraph summarizing summary of literature and research gap.

3. Authors need to state Originality of the article clearly.

4. There are a lot of interesting results, but discussions lack rigorousness. Authors needs to further strengthen this aspect.

5. The Figure 3 and 4 need to be explained more and add the scale on the figures.

6. Authors needs to add more clear pictures with visible scale for Figure 5, 6 and 7 also give more explanation of these pictures.

7. Authors needs to add color pictures for clarity for figure 8, 9 and 10.

8. To support the results in Figure 13~15, Authors needs to provide explanation while citing similar behavior from literature.

9. What was the model and specifications of the different equipment’s used for manufacturing and testing?

10. What was the source of materials used?

6. PLOS authors have the option to publish the peer review history of their article (what does this mean?). If published, this will include your full peer review and any attached files.

Reviewer #1: No

Reviewer #2: No

---

## [Author Response · Author response to Decision Letter 0]

16 Feb 2023

Response to reviewers’ comments–PONE-D-22-30437

(Manuscript Number: PONE-D-22-30437

Experimental study on Pore Structure Characteristics and Thermal Conductivity of Fibers Reinforced Foamed Concrete)

First of all, we are very grateful to the editors and reviewers for giving us the chance to revise the manuscript. The authors would like to acknowledge the reviewers for their valuable comments, which help us to improve the quality of this manuscript. We have revised the manuscript carefully according to the reviewers’ comments and suggestions. Changes have been highlighted in yellow or red in the revised manuscript. The manuscript has been revised according to the editor and reviewers’ comments. The response to the review comments is as follows.

Reviewer #1: Authors in this manuscript prepared the various concrete samples to evaluate their pore structure characteristics and thermal conductivity. But the manuscript needs to be improved for publishing in the journal.

1. The Abstract is so ambiguously written, and sentences are too long to understand. It was very hard for the reviewer to understand the combination of fibers and concrete just my reading the Abstract. Please rearrange the sentence structure along with clear explanation.

Response：The abstract has been revised as following:

ABSTRACT: The pore structure characteristics and thermal conductivity of foamed concrete (FC) reinforced with glass fibers (GF), polyvinyl alcohol fibers (PVAF) and polypropylene fibers (PPF) were investigated experimentally in this article. Firstly, GF, PVAF or PPF with different mass fractions (0%，1%，1.5% and 2%) were added to the Portland cement, fly ash and plant protein foaming agent to prepare the FC. Then, SEM tests, dry density tests, porosity tests, and thermal conductivity tests were carried out on FRFC. Later, the adhesion of GF, PVAF and FFF with different mass fractions to the cementitious base was investigated by SEM images of FRFC. Using Photoshop software and Image Pro Plus (IPP) software, the pore size distribution, shape factor and porosity of FRFC were analyzed. Finally, the effects of different mass fractions and lengths of the three types of fibers on the thermal conductivity of FRFC were discussed. The results indicated that appropriate mass fraction of fibers can play a role of refining small pores and separating large pores, improving the structural compactness, reducing the pore collapse phenomenon and optimizing the pore structure of FRFC. The three types of fibers can promote the optimization of cellular roundness and increase the proportion of small pores with diameters below 400 μm. The FC with larger porosity had smaller dry density. As the mass fraction of fibers increased, the thermal conductivity performed a phenomenon of first decrease and then increase. The three types of fibers with 1% mass fraction achieved relatively low thermal conductivity. Compared with the FC without fibers, the thermal conductivities of FRFC reinforced with 1% mass fraction of GF FC, PVAF were decreased by 20.73%, 18.23% and 7.00%, respectively.

2. Most important point is that you are considering the construction material but how the fibers concentration and their length are affecting mechanical properties? This should be the main concern first. Let’s suppose in your opinion the 1% has better results in thermal insulation but 2% has moderate results with improved mechanical performance. Please explain about this?

Response：It can be explained as following:

Foam concrete has larger porosity and light mass, so it is suitable for application in cast-in-place roof foam concrete insulation, foam concrete block, foam concrete exterior wall insulation board combined with foam concrete cast-in-place wall, etc. It can be seen that foam concrete is mainly used in the construction field for energy saving and thermal insulation, etc., which needs to have good thermal insulation performance, but should also meet the basic mechanical property requirements. The performance parameters of foam concrete are numerous. Among them, compressive strength and thermal conductivity are two particularly important parameters.

In this article, the effect of fibers on the pore structure and thermal conductivity of foam concrete were investigated. In addition, the overall compressive strength of foam concrete is low, roughly in the range of 0.5 ~ 10MPa, and it has been found that the appropriate amount and type of fibers can improve the pore structure and improve the mechanical properties of foam concrete. Therefore, the foam concrete reinforced with fibers prepared in this article not only meets the basic mechanical properties but also has high thermal conductivity. 

When polypropylene fibers were incorporated, their ultimate elongation was much better than that of cementitious materials. They can absorb part of the energy in the tensile process so that the "pseudo-ductility" phenomenon of fiber spanning at both ends of the initial cracks can be observed, which obviously alleviated the accelerated expansion of cracks, so that the flexural strength and deformation performance of foam concrete were obviously. Therefore, the flexural strength and deformation performance of foam concrete were improved. According to the specific applications of the prepared materials, the one with good thermal insulation properties was selected as optimal as possible while satisfying the mechanical properties.

3. Below Fig. 6, after the Fig. 7 explanation author has directly mentioned see Fig. 13 (a). It should be then placed after Fig. 7 as Fig. 8.

Response：The mistakes have been corrected in the revised manuscript.

4. Many term are unclear like in Figure 11, what does proportion mean? The author needs to explain this for better understanding.

Response: It has been revised as follows: In Table 2, the mass fraction mf has been explained, which is equal to fiber mass divided by the sum of cement mass and fly ash mass. For one type of FRFC in Fig.11, the proportion is between the number of pores with the same size and the total number of all pores with different sizes. The mass contents of the fibers in one type of FRFC were 0%, 1%, 1.5%, 2%, respectively. For one type of FRFC in Fig.12, the proportion is between the number of pore shape factor with the same value and the total number of all pores with different shape factors. The mass contents of the fibers in one type of FRFC were 0%, 1%, 1.5%, 2%, respectively. 

5. First author needs to explain the relation of proportion and pore size relation then address comment 6.

Response: It can be explained as follows:

For one type of FRFC in Fig.11, the proportion is between the number of pores with the same size and the total number of all pores with different sizes. The mass contents of the fibers in one type of FRFC were 0%, 1%, 1.5%, 2%, respectively. 

6. The results are simply explained which the reader can understand by seeing the results. Author need to explain with reason what phenomenon tend the proportion increase with PVAF when moving towards 2-4 micrometer.

Response: It can be explained as follows:

When the pore size was below 600 μm (relatively small), the proportion of FC reinforced with PVAF was high, which contains the pore size range of 200-400 μm. However, with the gradual increase in pore size, the proportion of the FC without fibers gradually increased, exceeding that of foam concrete with PVAF. This was mainly due to the fact that the addition of fibers can form a stable structure of three-dimensional network lap, which played the role of support and division and helped to form smaller and uniform size pores.

7. Authors need to add the discussion also in Result and Discussion section. Reviewer rarely can find any discussion to support the results.

Response: Discusson has been added in Section 3.4.

Section 3.4 Discussion

Fibers can effectively increase the proportion of small pores with diameters below 600 μm in the FC. The proportion of the shape factor in the range of 1 to 1.2 of the FC without fiber was 40%. After adding three types of fibers to the FC, the proportion of the shape factor in the range of 1 to 1.2 were greater than 40%. The changes in pore size and shape indicated that the fibers can effectively partition the air bubbles, making the pores smaller, closer in size, and more uniformly distributed, thus optimizing the roundness value and increasing the proportion of shape factor close to 1. Compared with the FC without fiber, the proportion of small pores with the diameters below 200 μm in FC can be increased by adding different types of fibers with different mass fractions. Fibers can inhibit the formation of a large number of large pores and was conducive to the formation of a uniform size pore structure, thus ensuring stable thermal insulation performance, which was consistent with the experimental test result that the appropriate mass friction of fibers can reduce the thermal conductivity obtained in this article (Compared with the FC without fibers, the thermal conductivity of the FC with the 1% mass friction of PPF, PVAF and GF was decreased by 20.73%, 18.23% and 7.00%, respectively).

From the aspect of thermal conductivity, the greater the porosity was, the smaller the average roundness value was, the smaller the thermal conductivity of FC was. The thermal conductivity of FC was affected by a variety of parameters such as porosity, the shape of the pore, etc., and cannot be analyzed simply with a certain parameter.

There were reports on the effect of fibers on the thermal conductivity of FC, see literature [35]. The relationship between the mass fraction of fibers and thermal conductivity was related to the type and length of fibers, and the construction process of FRFC. No completely consistent results have been obtained yet. But the appropriate mass friction of fibers is helpful to split large pores into small ones. The thermal conductivity of FRFC was decreased compared with the FC without fibers. However, with the increase of the mass fraction of fibers, the slurry liquidity decreased, and the fiber was easy to form clumps leading to poorer FC compatibility, especially the larger the fiber size clumping phenomenon was more obvious, but also in the mixing process would cause foam breakage. Therefore, the effect of fibers on thermal conductivity needs further comprehensive research. reports on the effect of fibers on the thermal conductivity of FC, see literature [35]. The relationship between the mass fraction of fibers and thermal conductivity was related to the type and length of fibers, and the construction process of FRFC. No completely consistent results have been obtained yet. But the appropriate mass friction of fibers is helpful to split large pores into small ones. The thermal conductivity of FRFC was decreased compared with the FC without fibers. However, with the increase of the mass fraction of fibers, the slurry liquidity decreased, and the fiber was easy to form clumps leading to poorer FC compatibility, especially the larger the fiber size clumping phenomenon was more obvious, but also in the mixing process would cause foam breakage. Therefore, the effect of fibers on thermal conductivity needs further comprehensive research.

8. In SEM images, different images have different resolutions. But when we are comparing the same factor in different combinations, we need to keep the resolution same. Also, you need to write the resolution separately as it is blur in images.

Response: It has been revised and the same resolution (0.7 nm) also can be seen clearly in Figs 5-7.

9. Whole English of the manuscript needs to be revised with the help of professional reviewer.

Response: The English has been improved.

Reviewer #2: The article titled “Experimental study on Pore Structure Characteristics and Thermal Conductivity of Fibers Reinforced Foamed Concrete” deals with an experimental investigation for effect of different fibers on the pore structure characteristics and thermal conductivity of foamed concrete (FC). Samples were manufactured by adding, glass fibers (GF), polyvinyl alcohol fibers (PVAF) and polypropylene fibers (PPF) with different volume contents (0%，1%，1.5% and 2%) to the Portland cement, fly ash and plant protein foaming agent.. The research is interesting, and methodology used is relevant. A clear materials and methods section are missing. Discussions lacks rigorousness and novelty is not clearly stated. Therefore, in current form, the article cannot be recommended for publication. It is recommended to accept the article subjected to following minor revisions.

1. Literature review is good but citation of articles on concrete made with different type of fibers as well as with different techniques are not many. Authors need to provide updated state of the art. They need to add some article reporting mechanical behavior of different fiber/fabric reinforced concretes with different techniques: for example:

Muhammad Imran Khan, Muhammad Umair, Khubab Shaker, Abdul Basit, Yasir Nawab & Muhammad Kashif (2020) Impact of waste fibers on the mechanical performance of concrete composites, The Journal of The Textile Institute, 111:11, 1632-1640,

DOI: 10.1080/00405000.2020.1736423

Umair, M., Khan, M.I., Nawab, Y. (2020). Green Fiber-Reinforced Concrete Composites. In: Kharissova, O., Martínez, L., Kharisov, B. (eds) Handbook of Nanomaterials and Nanocomposites for Energy and Environmental Applications. Springer, Cham. https://doi.org/10.1007/978-3-030-11155-7_113-1

Ali M, Khan MI, Masood F, Alsulami BT, Bouallegue B, Nawaz R, Fediuk R. Central composite design application in the optimization of the effect of waste foundry sand on concrete properties using RSM. InStructures 2022 Dec 1 (Vol. 46, pp. 1581-1594). Elsevier.

Response: It has been cited in the revised manuscript as follows.

Khan et al. investigate the effect of waste synthetic fibers (glass, polyester, and polypropylene) on the mechanical properties of fiber reinforced concrete (FRC) [25]. It was concluded that the glass fiber reinforced concrete has given the best performance with 4% fiber ratio in all aspects. Ali et al. used WFS as a partial replacement to reduce the use of fine aggregate in various concrete mixtures and to evaluate fresh concrete performance such as slump and mechanical properties such as compressive strength (CS), split tensile strength (STS), and flexural strength (FS). The results can be inferred that WFS can replace 20% of natural fine aggregate in order to obtain normal concrete strength. 

FC is mainly used in the construction field for energy saving and thermal insulation [27].

[25]Khan M I, Umair M, Shaker K, et al. Impact of waste fibers on the mechanical performance of concrete composites. The Journal of The Textile Institute, 2020, 111:11, 1632-1640.

[26] Ali M, Khan MI, Masood F, Alsulami BT, et al. Central composite design application in the optimization of the effect of waste foundry sand on concrete properties using RSM. Structures, 2022, 46:1581-1594.

[27] Umair M, Khan MI, Nawab Y. Green Fiber-Reinforced Concrete Composites. In: Kharissova, O., Martínez, L., Kharisov, B. (eds) Handbook of Nanomaterials and Nanocomposites for Energy and Environmental Applications. Springer, Cham, 2020.

2. Authors need to add one paragraph summarizing summary of literature and research gap.

Response: It has been added in Introduction as follows:

From above analysis, it can be found that FC is mainly used in the construction field for energy saving and thermal insulation. The performance parameters of foam concrete are numerous. Among them, compressive strength and thermal conductivity are two particularly important parameters. The overall compressive strength of FC is low. The appropriate mass friction and type of fibers can improve the pore structure and improve the mechanical properties of foam concrete. Therefore, the foam concrete reinforced with fibers prepared not only meets the basic mechanical properties but also has high thermal conductivity. When fiber is added into FC, the distribution, overlap and entanglement of fibers in the FC slurry affect the pore structure of FC. The pore structure is an important factor affecting the thermal conductivity of FC [28, 29]. However, there are few studies on the effects of different types of fibers on the pore structure characteristics and thermal conductivity of FC.

3. Authors need to state Originality of the article clearly.

Response: The Originality of the article has been clearly stated in Introduction as follows:

“…However, there are few studies on the effects of different types of fibers on the pore structure characteristics and thermal conductivity of FC.

With this aim, the pore structure characteristics and thermal conductivity of FC reinforced with glass fibers (GF), polyvinyl alcohol fibers (PVAF) and polypropylene fibers (PPF) were first prepared and tested in this article. Then, the adhesion of with different mass fractions of GF, PVAF and FFF to cementitious base was investigated by the SEM images of fibers reinforced FC (FRFC). The pore structure characteristic parameters of FRFC were analyzed using Photoshop software and Image Pro Plus (IPP) software. Finally, the effects of different mass fractions and lengths of three types of fibers on the thermal conductivity of FRFC were discussed”.

4. There are a lot of interesting results, but discussions lack rigorousness. Authors needs to further strengthen this aspect.

Response: Discussion has been added in Section 3.4.

Section 3.4 Discussion

Fibers can effectively increase the proportion of small pores with diameters below 600 μm in the FC. The proportion of the shape factor in the range of 1 to 1.2 of the FC without fiber was 40%. After adding three types of fibers to the FC, the proportion of the shape factor in the range of 1 to 1.2 were greater than 40%. The changes in pore size and shape indicated that the fibers can effectively partition the air bubbles, making the pores smaller, closer in size, and more uniformly distributed, thus optimizing the roundness value and increasing the proportion of shape factor close to 1. Compared with the FC without fiber, the proportion of small pores with the diameters below 200 μm in FC can be increased by adding different types of fibers with different mass fractions. Fibers can inhibit the formation of a large number of large pores and was conducive to the formation of a uniform size pore structure, thus ensuring stable thermal insulation performance, which was consistent with the experimental test result that the appropriate mass friction of fibers can reduce the thermal conductivity obtained in this article (Compared with the FC without fibers, the thermal conductivity of the FC with the 1% mass friction of PPF, PVAF and GF was decreased by 20.73%, 18.23% and 7.00%, respectively).

From the aspect of thermal conductivity, the greater the porosity was, the smaller the average roundness value was, the smaller the thermal conductivity of FC was. The thermal conductivity of FC was affected by a variety of parameters such as porosity, the shape of the pore, etc., and cannot be analyzed simply with a certain parameter.

There were reports on the effect of fibers on the thermal conductivity of FC, see literature [35]. The relationship between the mass fraction of fibers and thermal conductivity was related to the type and length of fibers, and the construction process of FRFC. No completely consistent results have been obtained yet. But the appropriate mass friction of fibers is helpful to split large pores into small ones. The thermal conductivity of FRFC was decreased compared with the FC without fibers. However, with the increase of the mass fraction of fibers, the slurry liquidity decreased, and the fiber was easy to form clumps leading to poorer FC compatibility, especially the larger the fiber size clumping phenomenon was more obvious, but also in the mixing process would cause foam breakage. Therefore, the effect of fibers on thermal conductivity needs further comprehensive research.

5. The Figure 3 and 4 need to be explained more and add the scale on the figures.

Response: It has been revised as follows: Fig. 3 gives the cross-section image of FRFC specimen. The cross-section size of the specimen was 100 mm × 100 mm. The scale on Fig. 3 was 0.5. The image without defects was selected as object, as shown in Fig. 4. The size of each image was 20 mm × 20 mm. The scale on Fig. 4 was 3.

6. Authors needs to add more clear pictures with visible scale for Figure 5, 6 and 7 also give more explanation of these pictures.

Response: More clear pictures with visible scale for Figs. 5-7 have been given. The SEM images in Figs. 5-7 have the same resolution (0.7 nm) and scale (100).

7. Authors needs to add color pictures for clarity for figure 8, 9 and 10.

Response: It can be explained as follows:

To improve the degree of contrast, binarization methods for black and white processing of the original photos are widely used. In this article, the original photos were binarized in black and white using image processing software (Photoshop software) to improve the clarity of the images. Then, the degree of binarization was adjusted by adjusting the threshold. Figs. 8, 9 and 10 show the binarization processed images. 

8. To support the results in Figure 13~15, Authors needs to provide explanation while citing similar behavior from literature.

Response: It can be explained as follows:

In Section 3.2.4, “…As the porosity increased, the smaller the proportion of solid phase in the FC, the dry density gradually decreased, which was consistent with that in literature [34].”

In Section 3.4, “There were reports on the effect of fibers on the thermal conductivity of FC, see literature [35]. The relationship between the mass fraction of fibers and thermal conductivity was related to the type and length of fibers, and the construction process of FRFC. No completely consistent results have been obtained yet. But the appropriate mass friction of fibers is helpful to split large pores into small ones. The thermal conductivity of FRFC was decreased compared with the FC without fibers. However, with the increase of the mass fraction of fibers, the slurry liquidity decreased, and the fiber was easy to form clumps leading to poorer FC compatibility, especially the larger the fiber size clumping phenomenon was more obvious, but also in the mixing process would cause foam breakage. Therefore, the effect of fibers on thermal conductivity needs further comprehensive research.”

9. What was the model and specifications of the different equipment’s used for manufacturing and testing?

Response: It has been given in Table 2.

10. What was the source of materials used?

Response: It has been given as follows:

The raw materials and their sources were as follows:

(1) Cement: P-O 42.5 grade ordinary silicate cement, purchased from a cement Co., Ltd. in Hainan, China.

(2) Fly ash: Grade I low-calcium fly ash, purchased from Chongqing Yuanheng Water Purification Material Factory in Chongqing, China.

(3) Foaming agent: using composite plant protein high-efficiency foaming agent, dilution times of 30 times, purchased from Qingdao Lego Environmental Protection Co., Ltd. in Qingdao, China.

(4) Water reducer: 540P type polycarboxylic acid water reducing agent (dilution ratio was 30 times), purchased from Shanghai Chenqi Chemical Technology Co., Ltd. in Shanghai, China.

(5) Fiber: glass fiber (GF), polyvinyl alcohol fiber (PVA), polypropylene fiber (PP), all purchased from Shanghai Chenqike Chemical Technology Co., Ltd. in Shanghai, China.

The types of fibers were GF (see Fig. 1(a), PVAF (see Fig. 1(b)) and PPF (see Fig. 1(c). The basic properties of the above three types of fibers are listed in Table 1.

---

## [Decision Letter · Decision Letter 1]

12 Jun 2023

Experimental study on Pore Structure Characteristics and Thermal Conductivity of Fibers Reinforced Foamed Concrete

PONE-D-22-30437R1

Dear Dr. Zhuang,

We’re pleased to inform you that your manuscript has been judged scientifically suitable for publication and will be formally accepted for publication once it meets all outstanding technical requirements.

Kind regards,

Yanping Yuan

Academic Editor

PLOS ONE

Additional Editor Comments (optional):

Reviewers' comments:

Reviewer's Responses to Questions

**Comments to the Author**

1. If the authors have adequately addressed your comments raised in a previous round of review and you feel that this manuscript is now acceptable for publication, you may indicate that here to bypass the “Comments to the Author” section, enter your conflict of interest statement in the “Confidential to Editor” section, and submit your "Accept" recommendation.

Reviewer #1: All comments have been addressed

Reviewer #3: All comments have been addressed

2. Is the manuscript technically sound, and do the data support the conclusions?

Reviewer #1: Yes

Reviewer #3: Yes

3. Has the statistical analysis been performed appropriately and rigorously? 

Reviewer #1: Yes

Reviewer #3: Yes

4. Have the authors made all data underlying the findings in their manuscript fully available?

Reviewer #1: Yes

Reviewer #3: Yes

5. Is the manuscript presented in an intelligible fashion and written in standard English?

Reviewer #1: Yes

Reviewer #3: Yes

6. Review Comments to the Author

Reviewer #1: (No Response)

Reviewer #3: the revision has been done well. the article can be accepted for publication in the plos one journal

7. PLOS authors have the option to publish the peer review history of their article (what does this mean?). If published, this will include your full peer review and any attached files.

Reviewer #1: No

Reviewer #3: No

---

## [Editor Report · Acceptance letter]

27 Jun 2023

PONE-D-22-30437R1 

Experimental study on Pore Structure Characteristics and Thermal Conductivity of Fibers Reinforced Foamed Concrete 

Dear Dr. Zhuang:

I'm pleased to inform you that your manuscript has been deemed suitable for publication in PLOS ONE. Congratulations! Your manuscript is now with our production department. 

Kind regards, 

on behalf of

Prof. Yanping Yuan 

Academic Editor

PLOS ONE